# A Comparison of Turning-Point Memories Among US and UK Emerging Adults: Adversity, Redemption, and Unresolved Trauma

**DOI:** 10.3390/bs15081127

**Published:** 2025-08-19

**Authors:** Cade D. Mansfield, Madisyn Carrington, Leigh A. Shaw

**Affiliations:** 1Department of Psychology, Franklin & Marshall College, Lancaster, PA 17603, USA; 2Department of Psychological Sciences, Weber State University, Ogden, UT 84408, USA; madisyncarrington@mail.weber.edu (M.C.); lshaw@weber.edu (L.A.S.)

**Keywords:** narrative identity, autobiographical memory, life story, psychological well-being, turning-point memories, redemption, meaning-making, emerging adulthood, culture

## Abstract

Turning-point memories, experiences that impact personal development, may be interpreted in ways that emphasize positive, negative, or mixed development because the memory prompt is open-ended with regard to event valence (i.e., it does not elicit ‘high’-point or ‘low’-point life events). Broadly, narratives that articulate how one has grown or changed for the better over time are positively associated with beneficial psychological characteristics and well-being, and are thought to be a cultural master narrative template in the United States (US). Recent work suggests cultural differences in the narration of adversity. Our mixed-methods study expands the literature on cultural comparisons of turning-point autobiographical memories by comparing themes in turning-point memory narratives of US and UK college-going emerging adults and by assessing whether or not narrative differences relate to changes in well-being and emotions after narration. Results suggest that turning points are characterized by memories of adversity and that redemptive narration is similar across samples in its frequency and associations with well-being and emotions. Discussion explores when and why redemptive narration may be beneficial for people from broad backgrounds.

## 1. Introduction

A turning point is an experience that shapes the self in an especially meaningful way ([33]; [36]). Similar definitions state that turning points are important life experiences that cause lasting alterations in a person’s developmental trajectory ([55]; [62]). Inasmuch as the experience matters by providing the source of ‘turning’, a tenet of narrative identity theory is the meanings made about the experience over time matter as much as the experience itself for shaping that trajectory. Supporting this idea are prospective longitudinal studies that point to narrative meaning-making as a causal force in people’s lives ([2]). For example, open exploration of negative experiences that focus on positive self-transformation predicts increased maturity and satisfaction with life over a nearly 10-year time period ([39]), themes of growth in narratives of personal transgressions predict increased self-compassion over time ([25]), and growth goals predict unique forms of positive functioning over a 3-year time period ([8]). Agency themes in narratives of client progress in therapy precede improvements in mental health ([1]), and newly recovering alcoholics whose narratives of their last drink featured redemption themes (positive self-transformation from negative beginnings) were more likely to be sober several months later than those who lacked redemption ([15]). Narrative processes and well-being correlates such as these exist within people within cultures, but how cultural differences contribute to narrative meaning-making has only recently begun to be studied (e.g., [57]). More work is required to gain a nuanced understanding of whether or not, when, and how culture may matter for narrative identity processes and its correlates. Our study compares turning-point memory narratives from emerging adults in a relatively homogenous US sample and a highly diverse UK sample to contribute to closing this gap in the literature.

In telling the story of a turning-point memory, people form and express interpretations of ‘me now’ that build from the story of ‘me as I was’ prior to the turning point ([35]). Turning-point memories are commonly collected as part of the Life Story Interview method ([27]). They are special among autobiographical memories in the sense that they are heterogenous: they may be events that happened to the person, over which the individual had no control, or the change may occur through volitional choice ([49]), and they are not necessarily positively nor negatively valenced. That is, compared to asking someone to narrate a high-point or a low-point memory from their life story, asking for a turning-point memory elicits a significant experience that may be positive, negative, or mixed in its emotional and developmental impact (they can be narrated as opening or closing opportunities), and participants recognize this ([47]). We focus our cross-national comparison on turning-point memories because they do not constrain participants to certain types of memories. Indeed, we anticipated that we might see the impacts of regional differences by exploring whether or not and how emerging adults in the two samples differed in narrative processing of turning points. We also explored turning-point memory narratives because past research shows that they are clearly identity salient—we learn and construct notions about who we are in telling these memories in ways that shape our understanding of the experience and the self ([36]; [18]; [58]). Self-report data show that they are rated as more central to a person’s life story, that is, the overall representation of how one became who one is over time, rather than simpler memories of transitions ([17]).

Because they represent substantial shifts in the self, turning-point memories may be especially important in emerging adulthood. Most emerging adults can identify having at least one turning point in their lives, and research shows that those memories may be a source of resilience for emerging adults who have faced adversity early in their lives ([55]; [13]). The salience of turning points makes sense for a period during which major developmental tasks include identity exploration, vocational pursuit, growth in romantic relationship experiences, and maturation of familial relationships ([4]; [36]). It is notable that those themes have been found in past research on turning-point memories with people in their fourth decade of life (36-year-olds, [48]). Constructing and sharing narratives of these especially self-relevant experiences may be one way to cope with the problem of identity that emerges during adolescence and exists as an issue in emerging adulthood and beyond ([40]; [5]).

### 1.1. Master Narratives and Well-Being

Master narratives are culturally valued, often invisible, and yet readily available templates for constructing and sharing the meaning of an experience. They may have important implications for the construction of turning-point narratives and narrative identity as well as the development and maintenance of personal well-being ([52], [53]). Indeed, they offer narrators a useful framework to make sense of life events in ways that listeners within that same culture, such as family members and loved ones, value and easily understand (e.g., [37]). To consciously, or unconsciously, draw on these templates when narrating a personal experience is to use the ‘normative, valued mode’ of meaning-making that is embedded in the framework of a culture.

Redemptive narration—when the bad is turned into good, challenge results in growth, stress in strength—is a master narrative template for storying challenging life experiences in the US where most narrative identity research has been conducted to date ([33]; [37]; [45]; but see also [57]). It is a widely studied hybrid construct in narrative identity work that taps elements of autobiographical reasoning and motivational and affective themes.

The hybridity of redemptive narration may make it especially suited for explaining perceived personal change that is based on reasoning about personal goals and emotional experiences ([38]). Redemptive themes emerge in a narrative when the protagonist describes an affectively negative event (e.g., parental divorce, personal loss, injury, achievement failure) as having had an affectively positive outcome (i.e., becoming resilient, learning to be patient, developing gratitude). [29]’ ([29]) foundational work on the redemptive self in the United States argues for six themes that express a redemptive self, including personal atonement (sin to salvation), upward mobility (rags to riches), enlightenment (ignorance to knowledge), psychological or moral development (immaturity to maturity), emancipation (bondage to freedom), and recovery (sickness to health). These themes make sense of how the person has become who they are today and showcase personal agency and commitments to others through generative concerns (e.g., [30]; [31]; [33]), which are ways of being a self that has been valued in the US culture. Redemptive themes structure the narrator’s remembered self as ongoing and embodying or moving toward positive functioning through past hardship. There is some payoff for this in US culture. Adults whose life stories overall feature redemptive imagery tend to score higher on life satisfaction, self-esteem, and personal coherence and lower on depression than adults who do not engage in redemptive narration ([33]). Furthermore, researchers have shown that US listeners like redemptive stories better than non-redemptive stories and they report liking people who tell redemptive stories better than they like people who tell non-redemptive stories ([21]; [38]). By expressing and fostering adaptive, culturally endorsed ways of being, redemptive narrative may support a person’s sense of well-being.

### 1.2. Redemptive Narratives of Difficult Life Events

Redemptive narration appears early in the lifespan, is present in turning-point memories, and seems to be associated with beneficial constructs. For example, adolescent turning-point memory narratives include redemption, and, as a scaled narrative variable, redemption is positively correlated with sophistication of meaning-making and self-esteem (among boys, [34]). Similarly, emerging adult turning-point memory narratives include redemption and it is positively associated with sophistication of meaning-making, and self and socially adaptive characteristics, such as optimism and generativity ([36]).

Outside of turning-point memory research per se, recent cross-cultural research that used both quantitative and qualitative coding on the narration of low points and life challenges, suggests that the prevalence of redemptive narration and its relationship with well-being measures varies across countries ([57]). Turner and colleagues’ work also suggested other ways of narrating adversity that was country-specific, such as acceptance of the difficulty rather than transformation of the self or experience through meaning-making. These findings echo a recent study conducted in the UK ([10]) that asked participants to narrate a national tragedy from the perspective of someone who experienced it directly and to narrate how one might be affected by that experience. Note that participants in the study did not directly experience the trauma. Although some redemption was present, results showed that UK citizens were just as likely to make meaning of trauma through affective themes of recuperation in their narratives, that is, moving forward from the event with lasting emotional scars without narrating it as a growth or positively transformative experience as redemptive narration does ([10]). However, this study’s findings do not directly address meaning-making in narratives of autobiographical memories because it did not assess autobiographical memories. Given past work on the presence and power of redemptively resolved autobiographical memory narratives, we explored whether or not emerging adults in our study, across nations, would engage in redemptive narration or some other form of meaning-making when sharing their own memories.

Overall, the sizeable literature on narration of life challenges broadly construed suggests individual, and potentially cultural, differences in the likelihood and impact of master narrative forms like redemptive narration. Notably, although it is considered a master narrative template, research within the United States has shown that people use redemption to narrate fewer than half of their important personal memories—even for stories about life’s difficulties ([14]; [24]; [60]). Hence even where the template is available, people can and do use alternative narrative forms to make meaning as described by [37] ([37]).

People from the US may be more likely than people of other nations to use redemptive narration because telling stories of personal transformation is embedded in the culture as a dominant master narrative. When people from the US use that narrative form, they may be especially likely to feel validated by others and positively about themselves because they are engaging with a normative and dominant form of meaning-making, and also because telling stories of positive transformation may actually make them feel better about themselves and the direction of their lives. People outside the US may have different master narrative templates at work that impact how they make meaning of personal challenges ([10]; [57]). If redemptive narration is not the ‘norm’ for people outside the US, they may not experience the same emotional or well-being boost that a US native would experience. Indeed, recent theory on the function of master narratives suggests that telling a narrative in a manner consistent with the master template may be beneficial because people feel validated and valued (e.g., [26]; [53]). If so, US emerging adults who tell redemptive narratives may report better well-being, more positive emotion, and less negative emotion than UK emerging adults given questions about whether or not redemption is a master narrative in the UK. That said, more research is needed to understand whether or not that is the case.

### 1.3. The Current Study

Turning-point memories are a reasonable, if not perfect, way to study implicit narrative processes, such as the role of broader dominant narrative templates on individual narration, because the prompt for the memory (see below) leaves a great deal of room for participant choice in the type of memory and how they express meaning about the memory. In line with these ideas, the current study set out to explore the following broad questions about turning-point memories in a US and UK sample.

(1)What themes characterize turning-point memories?(2)Do people in the two samples tell redemptive narratives with similar frequency?(3)Do they equally benefit from telling redemptive narratives?

We employed qualitative coding to gain a deeper understanding of the themes or primary ideas that recur in emerging adults’ turning-point memory narratives, expanding on similar ideas on coding redemption that have been expressed in past work ([45]) and to extend past work on the nature of turning-point memories across the lifespan ([62]; [48]). We also applied quantitative (i.e., scaled) coding systems to the turning-point memory narratives to analyze the extent to which individual differences in narrative features relate to aspects of personal well-being. To test the relationship between redemptive narration, well-being, and emotions of our participants, we collected well-being and emotional state measures before and after participants narrated turning points and used those to compute pre-narration to post-narration change in psychological state. Consistent with past work, we used a eudaimonic approach with Ryff’s measure of Psychological Well-being Scale (e.g., [2]; [9]; [25]). We also examined well-being from a more hedonic perspective by measuring people’s self-reported positive and negative emotional states. We used both qualitative and quantitative approaches to our data because the study resulted in a much smaller sample size for our UK participants. Overall, the current study sought to add to the growing literature on narrative identity processes across nations, which suggests that redemption in narratives of challenging events is not the “norm” for all cultures ([57]).

## 2. Materials and Method

The data for this study was from a broader study on turning-point narratives across cultures. Participants in the study were between 18 and 29 years of age, approximately the beginning and end of emerging adulthood. Although it is unclear exactly when emerging adulthood ends, we chose this age range following Arnett’s theory that the unique developmental characteristics of emerging adulthood, such as self-focus and identity exploration, may last throughout the 20s for many young people in industrialized nations (e.g., [4]; [6]) and because similar past studies on narrative identity and well-being used these ages (e.g., [59]). Participants were excluded if they failed to provide a codable narrative to the turning-point prompt. A codable narrative includes factual details such as what happened, who was involved, and interpretive mental state details describing the thoughts and feelings that caused people in the story to act or that resulted from the actions taken by people in the event ([41]; [43]). Or, as [12] ([12]) argued, narratives convey landscapes of action and consciousness. Non-codable responses to the narrative prompt did not include such detail, were made up of one sentence or less, were named but did not describe the turning point, or did not follow the prompt. Seventy-five percent of the emerging adult participants provided a codable narrative and were included in the analysis. The final sample consisted of 132 participants (*M*age = 20.73, *SD* = 2.83, 62.9% female). The UK sample (n = 25) was made up of emerging adults attending a regional university in the UK (*M*age = 20.32, *SD* = 2.19) and the US sample (n = 107) was made up of emerging adults attending a regional university in the Mountain West region of the United States (*M*age = 20.83, *SD* = 2.96). Frequency of female participants did not differ between the samples *X*^2^ (1, n = 132) = 0.02, *p* = 0.90. However, the samples differed in their stated nation of citizenship. While the US (>99%) and UK (74%) samples identified the US or UK/England, respectively, as their nation of citizenship, 26% of the UK sample identified other nations (e.g., Bulgaria, Greece). UK data collection was part of a student research project that was part of a study abroad experience. We made numerous efforts to recruit our sample and re-recruit but we had low participation rates overall. When the study abroad experience ended, we ceased data collection. This is discussed more in the study limitations. No generative AI was used in the creation of the study materials, in analyses, or in the development of the manuscript.

The nine-item version of the Psychological Well-being Scale ([50]) was used to measure 6 dimensions of psychological well-being: autonomy, environmental mastery, personal growth, positive relations with others, purpose in life, and self-acceptance. Items were rated on a 1 (strongly disagree) to 6 (strongly agree) Likert scale. This scale was collected before (α = 0.94) and after (α = 0.96) the turning-point narrative to assess differences in well-being in relation to differences in the relative presence of narrative features and by national context.

The Positive and Negative Affective Schedule (PANAS, [61]) was used to assess participants’ positive and negative emotions. Participants were asked to report on a 1 (very slightly or not at all) to 5 (extremely) Likert scale the extent to which they were experiencing 10 positively valenced emotions, with words like joy, enthusiasm, and happiness (α = 0.88), and 10 negatively valenced emotions, with words like fear, worry, and sadness (α = 0.88). Positive and negative emotions were rated before and after the turning-point narrative. At the *p* < 0.05 level (range 0.26–0.75), pre-narrative PANAS negative emotion items were positively correlated with each other, post-narrative PANAS negative emotions items were positively correlated with each other, pre-narrative PANAS positive emotion items were positively correlated with each other, and post-narrative PANAS positive emotion items were positively correlated with each other. Pre-narrative negative emotions items, the pre-narrative positive emotions items, the post-narrative negative emotion items, and the post-narrative positive emotion items were summed into four summary variables. We then computed pre- to post- change scores for positive emotions and negative emotions (see Table 1 for descriptives).

We used the well-established narrative turning-point prompt from The Life Story Interview ([27]) to elicit a turning-point narrative. Turning points were chosen because they are key autobiographical experiences that are open-ended with regard to event valence and significant to identity formation. The prompt is as follows:
“*In looking back over your life, it may be possible to identify certain key moments that stand out as turning points—episodes that marked an important change in you or your life story. Please identify a particular episode in your life story that you now see as a turning point in your life. If you cannot identify a key turning point that stands out clearly, please describe some event in your life wherein you went through an important change of some kind. For this event, please describe what happened, where and when, who was involved, and what you were thinking and feeling. Also, please say a word or two about what you think this event says about you as a person or about your life.*”

Participants used their laptops to access the Qualtrics survey. Participants provided consent on the first page of a Qualtrics survey. Those who consented were directed to the survey where they first completed a brief demographics section (e.g., age, gender) section and two measures not considered in the current study, The Big 5 Inventory and the Adult Attachment Inventory. Then, they completed the PANAS, the Psychological Well-being Scale, and read and responded to the turning-point prompt using a text box in Qualtrics (there were no space limitations for the narrative). After completing the narrative, participants again completed the PANAS and Psychological Well-being Scale and were thanked for their participation.

We followed standard coding procedures (e.g., [54]; [19]) using a commonly employed 5-point coding system for assessing redemptive narration, which is defined as a narrative that begins in a negative state and ends in a positive one ([38]). Contamination occurs when the reverse occurs: the narrative begins in a positive state and ends in a negative one. Coders were trained to reliably identify the beginning and ending states and the ending state was subtracted from the beginning state. All narratives were scored as one of the following −2, −1, 0, +1, +2 (positive scores = redemptive narration). Coders trained in weekly meetings by coding and then discussing codes for 10 to 12 narratives from an existing dataset of narratives (not analyzed or presented in this study). During practice, agreement and disagreement were discussed and disagreement was resolved through consensus by referring to the coding manuals. Practice continued until 80% agreement was reached on independently coded narratives. We formally assessed interrater reliability for redemption before proceeding to final independent coding on 20% of the sample narratives. Interrater reliability was good for redemption (*ICC* = 0.87).

## 3. Results

### 3.1. What Themes Characterize Turning-Point Memories?

In a bottom-up, initial pass through the turning-point narratives, we first noted that adversity stories seemed to predominate in the dataset for both samples. The turning-point memories that were shared with us expressed facing some kind of hardship. The context of the hardship varied from stories that were more expected in emerging adulthood, such as trying to get into college and being successful in challenging courses, struggling to live independently, and problems in intimate relationships, to stories that featured more severe hardship, such as becoming a teen mother, having a chronic or acute illness diagnosed (e.g., diabetes and leukemia), and death of loved ones. Given the timeframe of data collection stories of dealing with the global COVID-19 pandemic were also common. Out of the 132 narratives, 86% (114) were stories of memories about being shaped through adversity, broadly construed. The proportion of adversity narratives was high in emerging adults of both nations. In the US sample, 84% of narratives (90 out of 107) featured adversity and in the UK sample, 96% did (24 out of 25).

The third author subsequently used a thematic analysis approach to achieve a more nuanced assessment of the themes that young people express about their self and lives in these turning-point memories, which were predominantly set against a backdrop of adversity ([11]). Among the questions that [11] ([11]) say that qualitative researchers face, we knew that we wanted our analysis to result in rich descriptions of the main ways emerging adults described self-understanding/self-insight in their turning-point memories. Yet, we sought a parsimonious set of themes that captured rich descriptions. We continued to familiarize ourselves with the narrative data (phase 1) by reading through and discussing the narratives in several passes. Given the preponderance of adversity narratives, we read through them looking for expressions of positive meaning-making proposed by life story scholars ([28]) such as “sacrifice, recovery, growth, learning, improvement.” Hence from a thematic analysis perspective, we chose to use a theoretically driven approach in this phase after our initial inductive approach. This approach collapsed positively resolved narratives into two broad thematic camps ‘growth and learning’, which we decided was both overly simplistic and excessively overlapping. Thus, we went back into the narratives.

The third author went back and read all 132 narratives restarting phase 2 (generating initial codes) of thematic coding as she selected the sentence or sentences that expressed the primary insight that the narrator was taking away about themselves, their lives, or others. That is, if this is a turning-point memory, which phrase(s) captured how the person was changed? After reading through all of the narratives, she began to collate the extracted phrases into potential themes. We note that phases 3 (search for themes) and 4 (review of themes) of the [11] ([11]) thematic analysis technique were not exclusive because we periodically discussed the arising themes as the third author engaged in her search. We felt that this strengthened the analysis. However, the third author then gave names to the themes such that each name expressed the main ideas of the turning-point narratives as they related to and captured notions about the self. Again, consistent with a recursive qualitative approach to research ([11]; see also [20] in which we applied thematic analysis similarly), we went back and reviewed the themes and their meaning. The third author had identified 6 themes; strength, self-confidence, self-reliance, integrity/authenticity, lesson, enhanced connections with others. In an attempt to reduce overlap, we chose to collapse themes into the 3 psychological needs proposed by Self-Determination theory, Competence, Autonomy, and Relatedness (see below for a complete description), and a fourth category called unelaborated growth. Yet, we noted another group of narratives that did not end in positive resolution but that shared a common theme of ongoing negative impact from a negative event, which we named unresolved trauma. Hence, we developed the following final themes that captured main ideas about self in emerging adults’ positively and negatively resolved turning-point memories:

Mastery: Participant states that the turning point led to feelings of mastery and effectiveness in one’s activities, pride in their accomplishments, confidence (also lack of doubt) or security in their abilities, efficacy in their actions, achievement, or competence to achieve goals and/or difficult tasks.

Personal Growth/Autonomy: Participant states that the turning point led them to experience personal growth or transformation, a sense of self-acceptance, volition, will, psychological freedom, choice, decision-making that reflected their desires not necessarily what others expected or want them to do (not acting out of obligation), their choices express who they are, or they are acting authentically (doing what really interests them).

Relatedness: Participant states that the turning point led them to experience feelings of communion, relationship enhancement (they care about others and others care about them), connection to or closeness with others, or a lesson about how relationships should be.

Unelaborated Growth: Participant states that the turning point led them to grow and change in positive ways, but does not specify how. This category is expressed in narratives that lack adequate detail to determine another category and include references to vague or unelaborated growth.

Unresolved Trauma: Participant describes a negative event (e.g., failure, bullying, abuse, violence, loss and grief, betrayal) and the end of the narrative offers no resolution to the negative event. The overall idea is one of damage, harm, or ongoing hardship.

In collaboration with the first author, the third author developed a coding manual that explained each theme and trained two advanced undergraduates on the coding manual to identify themes through discussion and example coding. Once agreement was consistently high, they independently coded approximately 20% of the narratives to test the reliability of thematic coding. Average percentage agreement across the three coders was 75% and Cohen’s Kappa was adequate at 0.64. Each coder then proceeded to code the remaining narratives for themes and disagreement was resolved through discussion until consensus was reached. Figure 1 shows themes by sample.

#### Qualitative Examples of Themes Expressed

For emerging adults of both nations, turning-point memories are not only about adversity, but also are frequently about mastery experiences in the context of adversity and positive self-transformation in terms of personal growth. Yet, unresolved trauma was also a frequently present theme in these turning-point narratives. Unresolved trauma was a more likely theme in the UK than in the US sample, and it was the thematic category in which the samples differed the most. Qualitative analysis of turning-point memories reiterates the idea that we learn about ourselves from experiences that create “trouble” ([12]) and that for an important minority of our sample in both nations, the turning point is toward damage and diminishment. For example, this adversity turning point was coded as unresolved trauma:
*…I remember I was taking out the trash through the back and I had stopped to check my phone. I had a message from my best friend’s cousin saying that her boyfriend had hit her with a car. Obviously that’s not funny, but it’s so random so I call her from messenger and she’s incoherent. I knew. She wasn’t lying. My heart was pounding, and my stomach had done a complete 180. I felt like I couldn’t breathe. So I tell my supervisor, _____ (NAME), at the time and he lets me go. I leave and I drive to her house. To my best friend’s house. As I approached you can see the firetruck and cop cars. Her boyfriend went to leave after they had gotten in an argument and he backed up and hit her. She died. Just like that. 17 years old. She had her entire life ahead of her. He was 17 as well so you know what happened to him? Absolutely nothing. He’s out there free. This was a point in my life that broke me. I pushed people away. I cut everyone off. I didn’t want friends. The only person who I wanted to talk to was dead. I never got to say good bye. I never got to tell her how much I loved her or how much I appreciated her. The day she died something died within me too. It’s this gaping hole in my chest, and there is nothing I can do or anyone can do for that matter.*

Consistent with notions of stress-related growth ([56]), emerging adults in our sample most often shared memories with narrative themes of growth and enhancement as in this example:
*When I was in ______ (PLACE) visiting some family, we were on a hike and I had gone through a traumatic experience where I came close to losing my life. It was a wrong place and wrong time kind of deal. It was dangerous and scary. During the whole ordeal happening I was very calm and hopeful but accepting of what the bad outcome could be. This life or death situation as bad, scary, and traumatic it was, it taught me so much. I now look at life so differently, I cherish and present for the simple and little things in life. I know I have a purpose, and there is a reason I am still here today. I love life and everything about it a whole different way. I am very humbled and grateful because of it.*

Following calls from recent works, we explored themes in narratives told with a redemptive structure ([45]). We extended those calls by analyzing the themes by nation. We found that US participants were most likely to tell redemptive stories that focus on personal growth of the self, whereas UK participants did not elaborate deeply on how they grew, or they told stories of mastery that expressed their sense of competence and autonomy (see Figure 2).

### 3.2. Do People in the Two Samples Tell Redemptive Narratives with Similar Frequency?

Our next question focused on redemptive narration. Are people from the two samples similarly likely to move from negative to a positive ending in their turning-point memory narratives? Independent samples *t*-test with 1000 bootstrapped re-estimates showed that US and UK participants did not significantly differ on redemptive narration (see Table 1). Next, we analyzed the overall likelihood that narratives from the US and UK sample were redeemed. For this analysis, we created a dichotomous variable from redemption scores to explore whether or not redemption was more frequent as a proportion of narratives within each sample. Each participant with 1 or 2 on redemption was assigned a score of 1 on the dichotomous variable and those who scored anything else were assigned a 0. In the US sample, 71 out of 107 participants (66.4%) told a redemptive narrative. In the UK sample, 15 out of 25 participants (60%) did. There was not a significant difference in the proportion of turning-point memories told with a redemptive structure χ^2^ (1) = 0.36, *p* = 0.55. These findings do not support the conclusion that redemptive narration is more prevalent among a US than a UK sample of university going emerging adults. However, the quantitative findings should be considered as exploratory given the small UK sample size.

### 3.3. Do Emerging Adults in Our Samples Equally Benefit from Redemptive Narration?

The next set of quantitative analyses explored the extent to which redemptive narration might be associated with improved positive emotion, improved psychological well-being, and declining negative emotion after writing the narrative. First, to determine the feasibility of combining the pre- and post-PANAS items into single measures, we computed correlations among (1) the pre-narrative positive emotion PANAS items, (2) the post-narrative positive emotion PANAS items, (3) the pre-narrative negative emotion PANAS items, and (4) the post-narrative negative emotion PANAS items. Within each block of four sets of correlations, the emotion items were significantly positively correlated at the *p* < 0.05 level (*r*’s ranged from 0.26 to 0.75). We then computed four summary PANAS variables by summing across people’s emotion scores in each group.

The small UK sample size compromises our ability to do more complex analyses. But, to achieve a sense of the relationships that may exist between variability in redemptive narration and changes in emotional states and well-being, we performed simple bivariate correlations by sample. As shown in Figure 3, redemptive narration is associated with beneficial changes in emotional states and well-being regardless of nation. The correlations between redemption score and positive emotion change and redemption score and psychological well-being change are statistically significant in the US sample (*p* < 0.05). The other correlations do not reach statistical significance.

In both the US and UK samples, as redemptive scores increased, emerging adults reported increased positive emotion, increased well-being, and decreased negative emotions. The lack of statistical power in our UK sample precludes a formal statistical comparison of the strength of the correlations between the two samples, although the magnitudes of the correlations are quite similar. Regardless, our goal with the samples as they were was to achieve a sense of whether or not the pattern of relationships was similar, and they appear to be. The correlations are noteworthy in the context of the average change scores between the two samples in well-being, positive emotions, and negative emotions (Table 1). The correlations point to using a potential story form—redemption—as being potentially influential for feeling different after narrating a turning point. Overall, the pattern suggests that narrating turning points in a redemptive manner, rather than being from the US or not, is associated with beneficial well-being and psychological states. We explore potential meanings of these findings below.

## 4. Discussion

Emerging adulthood is a time of exploring and developing one’s identity. This is manifest in self-report data, but also in learning about the self by narrating memories that make up one’s life story. The present study suggests that turning-point memories, especially salient developmentally momentous experiences, are often about change through adversity. Moreover, our qualitative and quantitative analyses suggest that people in the US and people in a far more diverse, though smaller, UK sample use the redemptive narrative form in storying turning points. Our data open the door for ongoing studies to further examine the cultural contexts in which redemptive meaning-making provides benefits, as we discuss below.

### 4.1. Themes of Turning-Point Memories and Their Meanings for Emerging Adults

Thematically, our turning-point memory narratives were similar to past studies that showed that moments of maturation and realization are prominent themes ([55]). The mastery/competence and personal growth themes that were common in both samples in our study reflect this kind of maturation in emerging adulthood. We found that the majority of participants in both samples engaged in some degree of redemptive narration of turning points. These data suggest that UK emerging adults are not, on the whole, less likely to use redemptive themes when storying turning-point memories. These findings extend work on redemptive narration in the UK by showing that there may be differences in the likelihood of redeeming (rather than ‘recuperating’) when one tells a personally experienced negative memory as compared to when one imagines how others may tell a memory of a major trauma ([10]). When constructing meaning of personally experienced events that matter for identity, many people, regardless of national origin, may be pulled toward forming positive interpretations about the current self. Classic autobiographical memory experimental work on temporal self-appraisal with people from Western cultures suggests that we more positively appraise the current self if we feel that a failure is farther in the past (e.g., [46]; [63]). One possibility, perhaps especially for people from Western cultures, is that constructing narratives of positive self-transformation via redemption helps one feel subjectively distant from the past self that endured the adversity, thus providing the current self with a greater sense of strength or purpose and less vulnerability. The possibility that narrating might impact subjective distance from the past in ways that enhance the current self could be examined in future research on redemptive narration.

For US citizens, our data suggest that, perhaps aside from any subjective distance or self-enhancing bias, redemptive narration of negativity seems to be the cultural norm (e.g., [37]). Indeed, the overwhelming majority of US emerging adults used a redemptive narrative form in this study. [32] ([32]) suggested that redemptive narration is tied to the historical and cultural ethos of the United States in ways that have become intertwined with values like individualism, autonomy, and personal freedom. For our US emerging adults, we expected that narrating adversity turning points in the culturally dominant redemptive way would boost their positive emotions, lower their negative emotions, and perhaps increase their sense of well-being. Aligning a personal narrative with the ubiquitous cultural master narrative may benefit the self for a number of reasons that should be explored in ongoing research. As researchers have suggested, we do not yet know why redemptive narrating works as it does, when it does, or for whom (e.g., [26]). It may be beneficial because it feels normative or elicits more validation from a listener (e.g., [38]), and (or) because it might actually make one feel better to extract some sort of positive lesson or silver lining from a negative experience.

### 4.2. Turning-Point Memories as Opportunities for Eudaimonic Development

Importantly, our findings suggest that emerging adults in both the UK and US who told redemptive, as compared to those who did not tell redemptive, stories of adversity reported increased well-being after narration. We used the Ryff Psychological Well-being Scale, a well-known measure of eudaimonic well-being. The six domains assessed by this measure include the extent to which people feel that their life is rich in meaningful connection to others, they have control over self and environment, and they experience personal growth. We did not form a priori hypotheses about whether or not redemptive narration would foster increased eudaimonic well-being over such a short time frame in this study. However, it was associated with increased well-being. Perhaps narrating a turning-point memory that emphasizes the positive with themes of personal growth and mastery at the end of the story enhances one’s confidence in life going well and life being directed by the self rather than by forces out of one’s control. Likewise, redemptive narration may foster well-being because it requires emerging adults to construct a plausible account of how they have changed for the better, which may be broadly beneficial for eudaimonia (e.g., [7]). For example, the following redemptive narrative was written by an US emerging adult, and it creates a utility for the self in terms of becoming stronger and better over time. The narrator describes being diagnosed with diabetes, but soon after narrates gratitude for the formational experience, as it helped her become who she is today:
“*A turning point in my life was when I was diagnosed with type 1 diabetes almost 4 years ago at age 16…My mom and siblings were in the room when I was diagnosed, but my parents stayed in the hospital with me. I was terrified and went through a sort of grieving process over what could have been. Because of this event, however, I have grown immensely as a person. I have matured, become more patient, understanding, and have become more independent. Looking back on it, I’m grateful for this diagnosis, because it has helped me to become who I am today.*”

This story is an ideal example of a redemptive narrative for an emerging US adult. The protagonist takes a putatively negative, adverse experience—a diagnosis with a major metabolic disorder—and transforms it into a positive, meaningful event. Perhaps most importantly, in telling the story this way, the person shapes not only their understanding of the event, but also their sense of self ([35]). The retrospective construction is of a self that has become more patient, understanding, and independent. Below is an example of a narrative from an emerging UK adult with a growth-oriented redemptive ending that may be performing similar psychological work for the individual:
“*The key moment is a trip to a place called as ‘[TRAVEL DESTINATION]’ a travel destination in [NAME OF COUNTRY]. I went out for a trip with my friends who I trusted the most. When we were returning from the trip my friend got hit by a ‘Auto’. He was riding my bike and things turned out to be very bad. Thank God, he was not injured at all! We were a group of eight in that trip. But things just got out of hand. As this incident occurred in a small village, this made the people over there angry. We were surrounded by tens of villagers. This made us all very nervous. The thing that haunted me the most was that my family was unaware of this trip (I lied). Along with the fear of being surrounded by the villagers. My family was facing a severe loss in money at that time and that made me more sad. The most frustrating situation was that the friends who were with me on this trip leaved us alone in that situation. I have had no option but to inform this to my parents. Later, my father had arranged someone in that village to be stand besides me. The guy was a popular figure in that area. He solved the issue and the very next day, my father shows up even though the village was 100 miles from my home. The night of the accident, I was not able to sleep, thinking of what my friends did and what my family did during that situation. I have given a bigger place for my friends in my heart than my parents. But after this incident, my attitude has changed and currently my family is the most important people to me than anyone in this world. Now I am studying in [NAME OF COUNTRY] and leading a good life. I contact my parents everyday even though I am busy with my academics. Maybe this incident seems to be a silly one but influenced me to change my attitude towards the world.*”

Our results showed positive correlations between redemptive narration, post-narrative psychological well-being, and increases from pre-narration to post-narration in emerging adults in both nations. Although we are cautious in our interpretation because of the small UK sample size, these findings suggest that redemptive narration can be psychologically adaptive. Note that the featured emerging UK adult narrative tells a similar story of positive self-transformation via adversity as our featured emerging US adult narrative. Yet, the featured UK redemptive narrative is more communal as compared to the featured US redemptive narrative, which focuses on individuality and agency. Ongoing work examining redemptive narration will extend the field by not only examining the presence and absence of redemption in national samples, but also examining the frequency with which it takes different forms (see [45], for a developmental account of redemption forms).

Whereas scores of psychological well-being and positive emotionality were most likely to increase after writing a redemptive turning point, a finding that is consistent with past life story research ([51]), negative emotionality stayed the same. We believe that this may be due to the fact that adverse turning points are, at their core, riddled with negative emotions, making them a challenge to persevere through. Redemption may not be able to take away the negative feelings associated with adversity, but it can create positive meaning and positive emotions in individuals by weaving meaning and purpose into the experience through an overcoming of adversity, in addition to the pain and suffering endured.

Consistent with [2]’s ([2]) argument about the power of ‘living into the story’ one tells, there may be clinical implications of these findings for narrating far more traumatic memories. Redemptive meaning-making could be used as a tool to help those struggling to cope with adversity in their lives. Learning to reframe challenging experiences as redemptive may help people increase positive emotions by making sense of their experience (and associated negative emotions) in constructive, meaningful ways. Moreover, understanding that redemption might not necessarily reduce negative emotionality may allow the individual the space to experience their negative emotions while also becoming intentional about adapting to a new course of life because of the event. Redemption narratives may allow for the presence of negative emotions while bolstering positive emotionality that fosters long-term psychological well-being. Clinical intervention could be accomplished through cognitive restructuring that uses a redemptive framework, or possibly through narrative therapy focused on writing or creating a redemptive narrative to reframe challenging experiences ([44]). However, some of the turning points are thematically traumatic and as yet unresolved. They are likely important to the person’s sense of self and, as such, should be studied as authentic and perhaps valuable ways of grappling with adversity.

### 4.3. The Ongoing Story of Redemptive Narrative Research: One Size Is Unlikely to ‘Fit All’

Whether or not one can live with the trauma and narrate it in the life story in a psychologically adaptive way without positively storying that memory is an important question. Perhaps there is a way that people can accept and live with trauma without being defined by it that is psychologically beneficial that we have not measured in our studies. This idea is perhaps at the core of the potential benefits of recuperative narration ([10]), return identity themes in life stories ([45]) and “post” post-traumatic growth ([3]). These questions are especially relevant now as narrative studies are beginning to take account of cultural variability in meaning-making. Perlin and Fivush’s model may be especially useful in future work on cultural-situated meaning-making because its central focus is variability in how people express resolution. Indeed, they argue that “by attending to the situation and identity themes present across the entire narrative, we move beyond the specific forms of redemption that are culturally, religiously, and developmentally constrained ([45]).”

[45] ([45]) propose a contextualized, lifespan model of redemptive narration of adversity. They argue that there may be age-graded differences in how redemptive themes are manifest in adversity narratives and in the function redemptive themes serve. Their model begins with the type of adversity. They argue that we can have situation-specific adversity, such as failing a class in college, and identity-relevant adversity, such as struggling to choose a career path that makes sense for the self broadly. They go on to specify two forms of redemptive resolution (for other published works on resolution of negative experiences see [23]). One, return resolution, creates stability for the self by expressing a return of functioning, capability, or capacity that was challenged by adversity. The second, emergent resolution, signifies the development of a new strength, the identification of a novel skill, or a new insight about the self as a result of experiencing adversity (see also [42] for seminal theory on self-event connections in personal narratives). Hence in this model the way that resolution is expressed is critical for a nuanced understanding of redemptive narratives because it goes beyond quantitative approaches that show whether or not and the degree to which the individual redeemed to provide the quality of redemptive resolutions. Toward an even stronger refinement of redemptive narration theory, they suggest that return and emergent forms can express situation or identity themes. Taken as a whole this is an elegant and flexible model that we feel explains our turning-point data well. It suggests, for example, that a relatively minor situational adversity (e.g., failing a class) can be resolved with a return and identity-relevant theme such as rediscovering a love of learning that the narrator expresses as a core component of identity. Although we did not code our narratives in the same terms that Perlin and Fivush used, we recognize many of these types of stories in our turning-point dataset. In both the US and UK, broad themes of mastery, personal growth, and positive relatedness to others started out as situational adversity (for example, see the narrative about traveling in a new country with friends) but bloomed into stories of deeper identity concerns. That makes sense given that we prompted for turning points, which are identity-relevant, and our participants are emerging adults for whom identity development is important. Others began with an identity-relevant situation and resolved it with identity-relevant themes, such as the example above about being diagnosed with Type 1 Diabetes.

Yet overall, our work points to a tension in the progression of redemptive narrative research (see also [22]). For example, [38] ([38]) coded redemption as a structural element and argued that conceptually redemption has elements of both autobiographical reasoning (thinking about and making connections between self and experiences in narratives) and motivational and affective themes. Likewise, we coded redemption as a structural feature in our study and examined the themes in our narratives. We believe that this is valuable because it allows both quantitative and qualitative understanding of the data fitting the coding to the conceptualization. Further, it may be that the eudaemonic and hedonic benefits of redemptive telling arises out of a combination of structure (ending on a positive note), theme, and reasoning. As of now, this is an area for ongoing research ([26]).

### 4.4. Limitations and Considerations

The central limitation of this study was the UK sample size, which reduced cell size for comparisons and statistical power. Additionally, it is important to note that the UK sample was more diverse than the US sample in terms of where individuals identified their nation of origin. In the UK sample, although the majority identified the UK or England as their nation of citizenship (74%), about 1/4 identified nations such as Bulgaria, Greece, Nepal, Bangladesh, Qatar, and Spain. Hence, it is possible that this sample is not fully representative of emerging adults in the United Kingdom. The US sample was overwhelmingly born in the United States (>99%). In both samples, our data came from emerging adults attending college, who may not be representative of emerging adults more generally. Indeed, studying narrative identity development through the 30s in people who have neither attended nor completed 2- or 4-year degrees would be an illuminating area of future work. Finally, not a limitation, but a complication is that our thematic analysis work has much in common with qualitative content analysis ([16]), and we thank an anonymous reviewer for pointing this out. Whereas semantics (the words participants used) were critical in leading us to develop and apply themes, we also tried to understand the main ideas more deeply to draw inferences regarding the meanings they were expressing as themes of their turning points, and we believe the recursive process we used through the six phases is most in alignment with thematic analysis. Indeed, our intention was to apply the [11] ([11]) method, as no author had read about qualitative content analysis before the study. Having learned about both thematic analysis and qualitative content analysis, we anticipate future studies in narrative psychology benefiting by using each method.

## 5. Conclusions

Whereas redemptive narration has been frequently studied in the US, we know less about redemptive narration outside of the US (but see [57]). Inasmuch as the redemptive narrative is a cultural master narrative in the US, our findings suggest that it may also be a master narrative in the UK when emerging adults are narrating personally significant turning-point memories. This is an important addition to the current literature, which has, to date, considered redemptive narration to be a strong element of the US culture. Yet, future work with larger samples in the UK is required to fully address that question. Furthermore, our work suggests that redemptive narration is associated with short-term boosts to personal well-being and enhanced positively emotional states. Taken together, our findings suggest commonalities in how emerging adults across wide geographical distances and nations narrate turning-point memories. Although not appropriate for all memory retellings ([32]), redemptive narration may have positive implications for personal well-being. Indeed, our work suggests that turning-point experiences are not all remembered and told as resulting in a positive developmental trajectory. Formative traumatic experiences continue to influence identity, and ongoing research on living with those scars as part of the self, rather than transforming them, is especially important.

## Figures and Tables

**Figure 1 behavsci-15-01127-f001:**
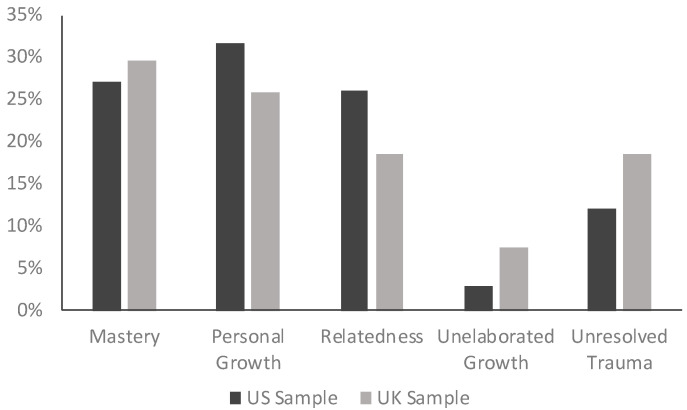
Percentage of themes by sample in all turning-point narratives.

**Figure 2 behavsci-15-01127-f002:**
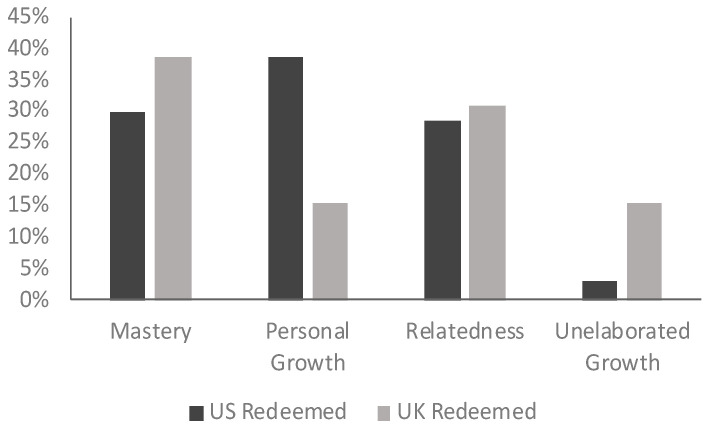
Turning-point narrative themes by sample in redemptive narratives.

**Figure 3 behavsci-15-01127-f003:**
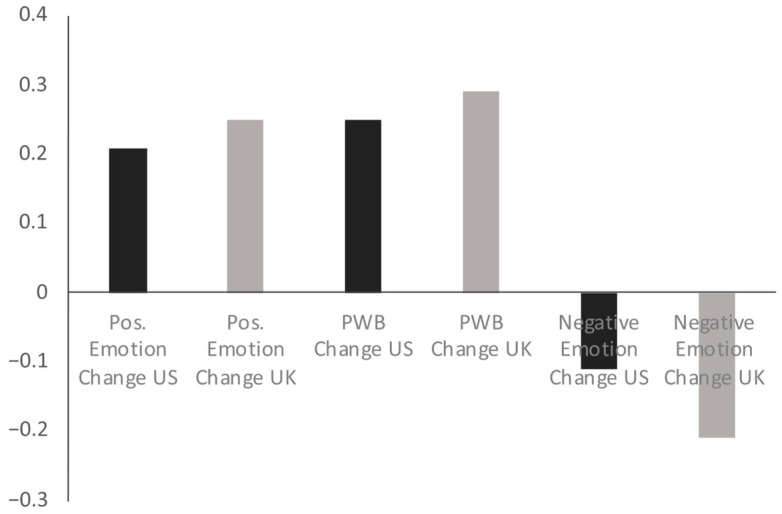
Correlations between redemptive narration score, positive emotion change, psychological well-being change, and negative emotion change scores.

**Table 1 behavsci-15-01127-t001:** Descriptives for Primary Variables.

	US (*M*, *SD*)	UK (*M*, *SD*)	Effect Size, *d* (95% CI)
Redemption Score	0.85 (0.80)	0.68 (0.75)	0.22 (−0.22, 0.65)
Pre-narrative Positive Emotion	29.15 (8.28)	26.67 (7.58)	0.24 (−0.19, 0.68)
Post-narrative Positive Emotion	31.48 (10.20)	28.79 (10.38)	0.20 (−0.23, 0.64)
Positive Emotion Change	2.33 (7.93)	2.13 (7.12)	0.01 (−0.42, 0.45)
Pre-narrative Negative Emotion	18.12 (6.86)	21.67 (8.87)	−0.43 (−0.87, 0.01)
Post-narrative Negative Emotion	17.74 (7.63)	22.50 (11.23)	−0.43 (−0.87, 0.01)
Negative Emotion Change	−0.39 (6.96)	0.83 (7.30)	−0.51 (−0.95, −0.07)
Pre-narrative Well-being *	4.27 (0.62)	3.75 (0.61)	0.69 (0.24, 1.13)
Post-narrative Well-being *	4.38 (0.70)	3.82 (0.68)	0.81 (0.36, 1.26)
Psych. Well-being Change	0.11 (0.28)	0.06 (0.29)	0.16 (−0.30, 0.59)

Note: Independent samples *t*-test shows that samples significantly differed on pre-narrative and post-narrative well-being two-tailed * = *p*s < 0.02. Required sample size for a difference that reaches a medium effect size (*d* = 0.05) with alpha = 0.05 and power at 0.8 G*power 3.1 indicates we would need a sample size of 208 with as few as 39 UK participants and 169 US participants. As an example of statistical power we actually achieved, post hoc achieved power for effect size 0.69 with alpha = 0.05, and n = 25 (UK) and n = 107 (US) power = 0.87 according to G*power 3.1.

## Data Availability

This study was not pre-registered. However, the authors agree to share the data from the study with interested colleagues. Please email the corresponding author for more information.

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
