# Peer review of "A Comparison of Turning-Point Memories Among US and UK Emerging Adults: Adversity, Redemption, and Unresolved Trauma"

_behavsci, 2025, doi:10.3390/bs15081127_

Round 1
Reviewer 1 Report
Comments and Suggestions for Authors
This study offers a valuable extension to the current research investigating how individuals narrate difficult life experiences outside of the US. The researchers made a strong case for their choice to analyse turning point narratives in emerging adulthood, and contextualised their study well in relation to past research by Blackie et al. (2023) that had examined similar research questions with hypothetical scenarios in the UK, rather than using personal autobiographical memories.
Despite the small heterogeneous UK sample limiting the generalisability of conclusions – which the researchers are themselves careful to acknowledge in their presentation and discussion of results – there is a place for this article. The results are interesting. It is noteworthy that the researchers found the frequencies of redemption narratives did not differ significantly between the US and UK and that the redemption narrative had the same well-being function in both samples. This study could provide a helpful foundation for future research in an area that does not often extend beyond North American samples.
I have some recommendations to help strengthen the manuscript:
- The researchers need to provide some justification for why emerging adulthood extends to the age of 29. The definition varies across researchers, but a justification with some relevant published research would be useful.
- The researchers explain in their introduction that they were interested in the extent to which turning point narratives were stressful events, and if so, the degree of redemption used in their narration. However, the finding that turning point narratives are often negative events seems to have come out organically out of their data analysis and it is interesting in its own right! Can the researchers find a way to frame their study rationale that is true to the process of discovery, rather than defining it as pre-defined hypothesis in the introduction?
- I had expected the researchers to code for recuperation as well, especially as they had a UK sample, but this was not done. A justification needs to be given for this decision, perhaps it is because of the sample UK sample size, or maybe another reason?
- The researchers say they did a reflective thematic analysis, but their process seems more akin to content coding to me. I’d encourage the researchers to review this paper on content coding - https://doi.org/10.1111/j.1365-2648.2007.04569.x - and reposition their results if they feel it more accurately reflects their analysis. If not, the researchers must consult the Braun & Clarke (2006) paper again and ensure they have followed their principles in their presentation of results. The write-up was missing some essential information for a thematic analysis.
- The discussion would benefit from a return to Perlin & Fivush (2021) to consider where the researchers’ themes fit in relation to this theory on the forms of redemption.
Author Response
A Comparison of Turning Point Memories among US and UK Emerging Adults: Adversity, Redemption, and Unresolved Trauma
Response to Reviewer 1’s Comments
|
||
Summary |
|
|
Thank you very much for your helpful comments and for taking the time to review our manuscript. We feel like your comments were particularly helpful in improving our work. Below, we provide your original comments (italicized in black) starting with the summary statement and any corresponding changes based on those comments (regular font in red). We include the page number, and the lines on which the manuscript revisions appear.
Reviewer 1 Summary Statement This study offers a valuable extension to the current research investigating how individuals narrate difficult life experiences outside of the US. The researchers made a strong case for their choice to analyse turning point narratives in emerging adulthood, and contextualised their study well in relation to past research by Blackie et al. (2023) that had examined similar research questions with hypothetical scenarios in the UK, rather than using personal autobiographical memories.
Despite the small heterogeneous UK sample limiting the generalisability of conclusions – which the researchers are themselves careful to acknowledge in their presentation and discussion of results – there is a place for this article. The results are interesting. It is noteworthy that the researchers found the frequencies of redemption narratives did not differ significantly between the US and UK and that the redemption narrative had the same well-being function in both samples. This study could provide a helpful foundation for future research in an area that does not often extend beyond North American samples.
I have some recommendations to help strengthen the manuscript:
|
||
Point-by-point response to Comments and Suggestions for Authors |
||
Comment 1: The researchers need to provide some justification for why emerging adulthood extends to the age of 29. The definition varies across researchers, but a justification with some relevant published research would be useful.
|
||
Response 1: That is a good catch. Thank you for pointing this out. We now explain our choice on page 5, lines 221 – 226, where we state, “Although it is unclear exactly when emerging adulthood ends, we chose this age range following Arnett’s theory that the unique developmental characteristics of emerging adulthood, such as self-focus and identity exploration, may last throughout the 20’s for many young people in industrialized nations (e.g., Arnett, 2000; Arnett et al., 2014) and because similar past studies on narrative identity and well-being used these ages (e.g., Waters & Fivush, 2015).”
|
||
Comment 2: The researchers explain in their introduction that they were interested in the extent to which turning point narratives were stressful events, and if so, the degree of redemption used in their narration. However, the finding that turning point narratives are often negative events seems to have come out organically out of their data analysis and it is interesting in its own right! Can the researchers find a way to frame their study rationale that is true to the process of discovery, rather than defining it as pre-defined hypothesis in the introduction?
|
||
Response 2: Thank you for reading carefully and for picking this up in our framing of the Introduction. We agree with your assessment. Because of that we have now removed this sentence from the Introduction “As we detail below, we were interested in the extent to which emerging adults of two nations (US and UK) would recall challenging stressful life memories as their turning point memories and whether or not they would differ in how they narrated those memories.” We anticipate that our presentation in the Introduction of why we chose to study turning points (i.e., that they can be positive or negative memories) including the information between lines 190 and 198 should sufficiently frame the study and leave in the ‘process of discovery’ that you mention. Lines 190-198 state: “Turning point memories are a reasonable, if not perfect, way to study implicit narrative processes, such as the role of broader dominant narrative templates on individual narration, because the prompt for the memory (see below) leaves a great deal of room for participant choice in the type of memory and how they express meaning about the memory. In line with these ideas, the current study set out to explore the following broad questions about turning point memories in a US and UK sample. 1) What themes characterize turning point memories? 2) Do people in the two samples tell redemptive narratives with similar frequency? 3) Do they equally benefit from telling redemptive narratives?”
Comment 3: I had expected the researchers to code for recuperation as well, especially as they had a UK sample, but this was not done. A justification needs to be given for this decision, perhaps it is because of the sample UK sample size, or maybe another reason?
Response 3: We agree that our treatment of recuperation in the Introduction (approx. top of page 4) would lead to the expectation that we coded for recuperation. But we did not do so, which is confusing for the reader. To ameliorate this, we changed the sentence below to broaden the coding we did. However, we kept the recuperation content because it is relevant and recuperation fits with the ‘return’ themes proposed by Perlin and Fivush. Page 4 (lines 156-161) now reads: However, this study’s findings do not directly address meaning-making in narratives of autobiographical memories because it did not assess autobiographical memories. Given past work on the presence and power of redemptively resolved autobiographical memory narratives, we explored whether or not emerging adults in our study, across nations, would engage in redemptive narration or some other form of meaning-making when sharing their own memories.
Comment 4: The researchers say they did a reflective thematic analysis, but their process seems more akin to content coding to me. I’d encourage the researchers to review this paper on content coding - https://doi.org/10.1111/j.1365-2648.2007.04569.x - and reposition their results if they feel it more accurately reflects their analysis. If not, the researchers must consult the Braun & Clarke (2006) paper again and ensure they have followed their principles in their presentation of results. The write-up was missing some essential information for a thematic analysis.
Response 4: This is a very important comment for us. Thank you for the useful reference and suggestion. It led us to think more carefully about how we explained our analysis. None of the authors were familiar with content coding when doing our analyses but we are very happy to understand that technique and its commonalities and differences with thematic analysis. We have rewritten two sections of the paper to better explain our thematic analysis (Braun & Clarke, 2006) approach. We feel like the confusion on our technique in the original paper resulted from not fully explaining all of the steps we took. That is, in trying to be simple and clear, we left out major details. We have those in the paper now. Lines 330-396 on pages 8 and 9 now read: “The third author subsequently used a thematic analysis approach to get a more nuanced assessment of the themes that young people express about their self and lives in these turning point memories, which were predominantly set against a backdrop of adversity (Braun & Clark, 2006). Among the questions that Braun and Clarke (2006) say that qualitative researchers face, we knew that we wanted our analysis to result in rich descriptions of the main ways emerging adults described self-understanding/self-insight in their turning point memories. Yet, we sought a parsimonious set of themes that captured rich descriptions. We continued to familiarize ourself with the narrative data (phase 1) by reading through and discussing the narratives in several passes. Given the preponderance of adversity narratives, we read through them looking for expressions of positive meaning-making proposed by life story scholars (McAdams, 2001) such as “sacrifice, recovery, growth, learning, improvement”. Hence from a thematic analysis perspective, we chose to use a theoretically driven approach in this phase after our initial inductive approach. This approach collapsed positively resolved narratives into two broad thematic camps ‘growth and learning’, which we decided was both overly simplistic and excessively overlapping. Thus, we went back into the narratives. The third author went back and read all 132 narratives restarting phase 2 (generating initial codes) of thematic coding as she selected the sentence or sentences that expressed the primary insight that the narrator was taking away about themselves, their lives, or others. That is, if this is a turning point memory, which phrase(s) captured how the person was changed? After reading through all of the narratives, she began to collate the extracted phrases into potential themes. We note that phases 3 (search for themes) and 4 (review of themes) of the Braun and Clarke (2006) thematic analysis technique were not exclusive because we periodically discussed the arising themes as the third author engaged in her search. We felt that this strengthened the analysis. However, the third author then gave names to the themes such that each name expressed the main ideas of the turning point narratives as they related to and captured notions about the self. Again, consistent with a recursive qualitative approach to research (Braun & Clarke, 2006, see also Grysman et al., 2024 in which we applied thematic analysis similarly), we went back and reviewed the themes and their meaning. The third author had identified 6 themes; strength, self-confidence, self-reliance, integrity/authenticity, lesson, enhanced connections with others. In an attempt to reduce overlap, we chose to collapse themes into the 3 psychological needs proposed by Self-Determination theory, Competence, Autonomy, and Relatedness (see below for a complete description), and a fourth category called unelaborated growth. Yet, we noted another group of narratives that did not end in positive resolution but that shared a common theme of ongoing negative impact from a negative event, which we named unresolved trauma. Hence, we developed the following final themes that captured main ideas about self in emerging adults’ positively and negatively resolved turning-point memories: Mastery: Participant states that the turning point led to feelings of mastery and effectiveness in one’s activities, pride in their accomplishments, confidence (also lack of doubt) or security in their abilities, efficacy in their actions, achievement, or competence to achieve goals and/or difficult tasks. Personal Growth / Autonomy: Participant states that the turning point led them to experience personal growth or transformation, a sense of self-acceptance, volition, will, psychological freedom, choice, decision-making that reflected their desires not necessarily what others expected or want them to do (not acting out of obligation), their choices express who they are, or they are acting authentically (doing what really interests them). Relatedness: Participant states that the turning point led them to experience feelings of communion, relationship enhancement (they care about others and others care about them), connection to or closeness with others, or a lesson about how relationships should be. Unelaborated Growth: Participant states that the turning point led them to grow and change in positive ways, but does not specify how. This category is expressed in narratives that lack adequate detail to determine another category and include references to vague or unelaborated growth. Unresolved Trauma: Participant describes a negative event (e.g., failure, bullying, abuse, violence, loss and grief, betrayal) and the end of the narrative offers no resolution to the negative event. The overall idea is one of damage, harm, or ongoing hardship. In collaboration with the first author, the third author developed a coding manual that explained each theme and she trained two advanced undergraduates on the coding manual to identify themes through discussion and example coding. Once agreement was consistently high, they independently coded approximately 20% of the narratives to test the reliability of thematic coding. Average percentage agreement across the three coders was 75% and Cohen’s Kappa was adequate at .64. Each coder then proceeded to code the remaining narratives for themes and disagreement was resolved through discussion until consensus was reached. Figure 1 shows themes by sample.”
And lines 707 through 717 on page 16 read as follows: “Finally, not a limitation, but a complication is that our thematic analysis work has much in common with qualitative content analysis (Elo & Kyngäs, 2007), and we thank an anonymous reviewer for pointing this out. Whereas semantics (the words participants used) were critical in leading us to develop and apply themes, we also tried to understand the main ideas more deeply to draw inferences regarding the meanings they were expressing as themes of their turning points and we believe the recursive process we used through the 6 phases is most in alignment with thematic analysis. Indeed, our intention was to apply the Braun and Clarke (2006) method as no author had read about qualitative content analysis before the study. Having learned about both thematic analysis and qualitative content analysis, we anticipate future studies in narrative psychology benefiting by using each method.”
Comment 5: The discussion would benefit from a return to Perlin & Fivush (2021) to consider where the researchers’ themes fit in relation to this theory on the forms of redemption.
Response 5: We agree, thank you for pointing this out. To address this, we made substantial changes to lines 639-693 on pages 14 through 15 which now reads as: Whether or not one can live with the trauma and narrate it in the life story in a psychologically adaptive way without positively storying that memory is an important question. Perhaps there is a way that people can accept and live with trauma without being defined by it that is psychologically beneficial that we have not measured in our studies. This idea is perhaps at the core of the potential benefits of recuperative narration (Blackie et al., 2023), return identity themes in life stories (Perlin & Fivush, 2021) and “post” post-traumatic growth (Adler & Schwaba, 2024). These questions are especially relevant now as narrative studies are beginning to take account of cultural variability in meaning-making. Perlin and Fivush’s model may be especially useful in future work on cultural-situated meaning-making because its central focus is variability in how people express resolution. Indeed, they argue that “by attending to the situation and identity themes present across the entire narrative, we move beyond the specific forms of redemption that are culturally, religiously, and developmentally constrained (Perlin & Fivush, 2021, p. 28).” Perlin & Fivush (2021) propose a contextualized, lifespan model of redemptive narration of adversity. They argue that there may be age-graded differences in how redemptive themes are manifest in adversity narratives and in the function redemptive themes serve. Their model begins with the type of adversity. They argue that we can have situation-specific adversity, such as failing a class in college, and identity-relevant adversity, such as struggling to choose a career path that makes sense for the self broadly. They go on to specify two forms of redemptive resolution (for other published works on resolution of negative experiences see Mansfield et al., 2010). One, return resolution, creates stability for the self by expressing a return of functioning, or capability, or capacity that was challenged by the adversity. The second, emergent resolution, signifies the development of a new strength, the identification of a novel skill, or a new insight about the self as a result of experiencing the adversity (see also Pasupathi, et al., 2007 for seminal theory on self-event connections in personal narratives). Hence, in this model, the way that resolution is expressed is critical for a nuanced understanding of redemptive narratives because it goes beyond quantitative approaches that show whether or not and the degree to which the individual redeemed to provide the quality of redemptive resolutions. Toward an even stronger refinement of redemptive narration theory, they suggest that return and emergent forms can express situation or identity themes. Taken as a whole, this is an elegant and flexible model that we feel explains our turning point data well. It suggests, for example, that a relatively minor situational adversity (e.g., failing a class) can be resolved with a return and identity relevant theme such as rediscovering a love of learning that the narrator expresses as a core component of identity. Although we did not code our narratives in the same terms that Perlin and Fivush used, we recognize many of these types of stories in our turning point dataset. In both the US and UK, broad themes of mastery, personal growth, and positive relatedness to others started out as situational adversity (for example, see the narrative about traveling in a new country with friends) but bloomed into stories of deeper identity concerns. That makes sense given that we prompted for turning points, which are identity relevant, and our participants are emerging adults for whom identity development is important. Others began with an identity-relevant situation and resolved it with identity relevant themes, such as the example above about being diagnosed with Type 1 Diabetes. Yet overall, our work points to a tension in the progression of redemptive narrative research (see also Mansfield et al., 2025). For example, McLean and colleagues (2020) coded redemption as a structural element and argued that conceptually redemption has elements of both autobiographical reasoning (thinking about and making connections between self and experiences in narratives) and motivational and affective themes. Likewise, we coded redemption as a structural feature in our study and examined the themes in our narratives. We believe that this is valuable because it allows both quantitative and qualitative understanding of the data fitting the coding to the conceptualization. Further, it may be that the eudaemonic and hedonic benefits of redemptive telling arise out of a combination of structure (ending on a positive note), theme, and reasoning. As of now, this is an area for ongoing research (Mansfield et al., 2023).
|
Reviewer 2 Report
Comments and Suggestions for Authors
In one study, researchers compared a US and UK sample of young adults, collecting turning point memories from the samples along with measures of well-being before and after the memory collection. Researchers used a mixed method approach, finding similar redemption themes of mastery, growth, relatedness, and small numbers of unresolved trauma in both samples. US sample was more likely to discuss their personal growth in detail, while UK sample was more likely to discuss mastery and autonomy. Positive associations with well-being were found for the US sample. Overall patterns were similar for both samples, indicating shared patterns across different cultures.
The addition of a non-US sample to this line of research is welcome, and I appreciate the lengthy amount of time and effort that goes into the qualitative coding of such turning point memories. I think this research does add new information to the field, but it is quite limited in its scope and impact due to its very small, UK sample size. With a fully powered UK sample, this could be much more impactful work, as the theoretical background, design and methodology are sound. Overall, I am left wondering why the authors did not collect a larger sample or follow up this first step with a study 2, using better matched pair of samples from the US and UK?
Below are additional minor comments and requests for the manuscript, in order of appearance, not importance.
-I think the organization of the manuscript would be improved with additional subheadings. In the intro, page 3 line 91 and page 4, line 186 seem to be natural places to add sub headers that would signal to the reader changes in the topic of the intro.
Ditto for the materials and method section as well—the organization is good, but clarity would increase with subheadings. Especially around line 358, I began to wonder if the examples of narratives weren’t better for the discussion or in a separate table, but I think simply having clear subheadings could help better steer the reader through the sections of method/materials.
-can the authors include a power analysis of the two samples? The UK sample is underpowered, but it would be good for the readers to know about the US sample, and what size should be needed for the UK.
-Also please describe in the method why data collection ended. Without an a priori power analysis to determine sample size, some other methodology must have been used to conclude the collection of data—please describe. This would also be a good place to describe to readers why more participants weren’t collected from the UK sample—what were the constraints?
-line 259, first table. The change in positive emotion is interesting, and it’s in line with previous research showing that state well-being levels improve after life story recall (though it is a short-lived boost)
https://doi.org/10.1111/jopy.12449
-page 8, line 356. Are there statistics to back up this claim? Please reference.
-line 400. I would adjust the language to present this finding as it is misleading as written. The difference between the two samples was not significant. “A small difference” indicates a significant finding with a small effect size.
-Please include the effect sizes for the t-tests.
-line 476—I’m not sure that research with East Asian samples supports this idea of distance from negative memories. Suggestion to adjust the scope of that statement.
-line 554. Clarification here: is there research suggesting that a redemption arc is not culturally dominant in the UK? This statement indicates that we know what the culturally dominant narrative in the UK is, and that it is NOT redemption arcs. and I don’t think that’s totally true, otherwise your paper wouldn’t be taking the first steps to investigate it. Suggestion to end the sentence at “psychologically adaptive”.
Very minor grammar finds: lines 389 and 390 switch into a different verb tense rather than the past tense. “extended”, “found” “were”.
Author Response
A Comparison of Turning Point Memories among US and UK Emerging Adults: Adversity, Redemption, and Unresolved Trauma
Response to Reviewer 2’s Comments
|
||
Summary |
|
|
Thank you very much for your helpful comment and for taking the time to review our manuscript. We feel like your comments were particularly helpful in improving our work. Below, we provide your original comments (italicized in black) starting with the summary statement and any corresponding changes to the revised manuscript based on those comments (regular font in red). We include the page number, and the lines on which revisions appear.
Reviewer 1 Summary Statement In one study, researchers compared a US and UK sample of young adults, collecting turning point memories from the samples along with measures of well-being before and after the memory collection. Researchers used a mixed method approach, finding similar redemption themes of mastery, growth, relatedness, and small numbers of unresolved trauma in both samples. US sample was more likely to discuss their personal growth in detail, while UK sample was more likely to discuss mastery and autonomy. Positive associations with well-being were found for the US sample. Overall patterns were similar for both samples, indicating shared patterns across different cultures. The addition of a non-US sample to this line of research is welcome, and I appreciate the lengthy amount of time and effort that goes into the qualitative coding of such turning point memories. I think this research does add new information to the field, but it is quite limited in its scope and impact due to its very small, UK sample size. With a fully powered UK sample, this could be much more impactful work, as the theoretical background, design and methodology are sound. Overall, I am left wondering why the authors did not collect a larger sample or follow up this first step with a study 2, using better matched pair of samples from the US and UK?
Below are additional minor comments and requests for the manuscript, in order of appearance, not importance.
|
||
Point-by-point response to Comments and Suggestions for Authors |
||
Comment 1: I think the organization of the manuscript would be improved with additional subheadings. In the intro, page 3 line 91 and page 4, line 186 seem to be natural places to add sub headers that would signal to the reader changes in the topic of the intro. Ditto for the materials and method section as well—the organization is good, but clarity would increase with subheadings. Especially around line 358, I began to wonder if the examples of narratives weren’t better for the discussion or in a separate table, but I think simply having clear subheadings could help better steer the reader through the sections of method/materials.
|
||
Response 1: Thank you for pointing this out. We agree with this comment. In fact, prior to reading the journal author guidelines, which seemed to suggest fewer subheadings, we had more subheadings in the paper. We have now added more subheadings throughout to help clean up the flow of ideas and provide signposts to the reader.
Note inclusion of new subheadings at the following lines and pages: 1.1 Master Narratives and Well-being, line 92, page 3 1.2 Redemptive Narratives of Difficult Life Events, line 133, page 3 1.3 The Current Study, line 189, page 4 3.1.1 Qualitative Examples of Themes Expressed, line 399, page 9 4.1 Themes of Turning Point Memories and their Meanings for Emerging Adults, line 505, page 12 4.2 Turning Point Memories as Opportunities for Eudaimonic Development, line 545, page 14 4.3 The Ongoing Story of Redemptive Narrative Research: One Size is Unlikely to ‘Fit All’. Line 637 p. 14 4.4 Limitations and Considerations, line 695, p. 16
|
||
Comment 2: can the authors include a power analysis of the two samples? The UK sample is underpowered, but it would be good for the readers to know about the US sample, and what size should be needed for the UK.
|
||
Response 2: Yes, thank you for this idea. We now include this information in a note below Table 1 (where the comparisons occur). That note reads: Required sample size for a difference that reaches a medium effect size (d = .05) with alpha =.05 and power at .8 G*power 3.1 indicates we would need a sample size of 208 with as few as 39 UK participants and 169 US participants. As an example of statistical power we actually achieved, post-hoc achieved power for effect size .69 with alpha = .05, and n = 25 (UK) and n = 107 (US) power = .87 according to G*power 3.1.
Comment 3: Also please describe in the method why data collection ended. Without an a priori power analysis to determine sample size, some other methodology must have been used to conclude the collection of data—please describe. This would also be a good place to describe to readers why more participants weren’t collected from the UK sample—what were the constraints?
Response 3: Agreed, thanks, this is an important point to make. We describe this now in the method section as suggested. On Line 245 – 247, page 6 we now state: UK data collection was part of a student research project that was part of a study abroad experience. We made numerous efforts to recruit our sample and re-recruit but we had low participation rates overall. When the study abroad experience ended, we ceased data collection. This is discussed more in the study limitations.
Comment 4: line 259, first table. The change in positive emotion is interesting, and it’s in line with previous research showing that state well-being levels improve after life story recall (though it is a short-lived boost) https://doi.org/10.1111/jopy.12449
Response 4: Great pick-up, thanks! We now acknowledge this finding in our Discussion section on lines 610-612, page 14 where we note: Whereas scores of psychological well-being and positive emotionality were most likely to increase after writing a redemptive turning point, a finding that is consistent with past life story research (Steiner et al., 2019), negative emotionality stayed the same.
Comment 5: page 8, line 356. Are there statistics to back up this claim? Please reference.
Response 5: We were probably speculating too much there. In general, I was thinking about negative meaning-making about the self negatively correlating with positive outcomes and positively correlating with negative ones (e.g. Banks & Salmon, 2013). But it was likely too much of a reach as written. So, page 9 line 402 now sticks more closely to our findings only by stating: “Yet, unresolved trauma was also a frequently present theme in these turning point narratives.”
Comment 6: line 400. I would adjust the language to present this finding as it is misleading as written. The difference between the two samples was not significant. “A small difference” indicates a significant finding with a small effect size.
Response 6: Thank you for this comment. We have revised that section. It now reads (lines 442-447, page 10): “Do people in the two samples tell redemptive narratives with similar frequency? Our next question focused on redemptive narration. Are people from the two samples similarly likely to move from negative to a positive ending in their turning point memory narratives? Independent samples t-test with 1,000 bootstrapped re-estimates showed that US and UK participants did not significantly differ on redemptive narration (see Table 1).”
Comment 7: Please include the effect sizes for the t-tests.
Response 7: Absolutely. We have now added point estimate of Cohen’s d and 95% CI to Table 1, page 6. Thank you for pointing this out.
Comment 8: line 476—I’m not sure that research with East Asian samples supports this idea of distance from negative memories. Suggestion to adjust the scope of that statement.
Response 8: We have limited the scope of these statements as follows (lines 475-484 page 12): “Classic autobiographical memory experimental work on temporal self-appraisal with people from Western cultures suggests that we more positively appraise the current self if we feel that a failure is farther in the past (e.g., Ross & Wilson, 2003; Wilson & Ross, 2001). One possibility, perhaps especially for people from Western cultures, is that constructing narratives of positive self-transformation via redemption helps one feel subjectively distant from the past self that endured the adversity, thus providing the current self with a greater sense of strength or purpose and less vulnerability. The possibility that narrating might impact subjective distance from the past in ways that enhance the current self could be examined in future research on redemptive narration.”
Comment 9: line 554. Clarification here: is there research suggesting that a redemption arc is not culturally dominant in the UK? This statement indicates that we know what the culturally dominant narrative in the UK is, and that it is NOT redemption arcs. and I don’t think that’s totally true, otherwise your paper wouldn’t be taking the first steps to investigate it. Suggestion to end the sentence at “psychologically adaptive”.
Response 9: Agreed, good point here. We now end the sentence with psychologically adaptive and the sentence is now at lines 600 to 602 on page 14. “Although we are cautious in our interpretation because of the small UK sample size, these findings suggest that redemptive narration can be psychologically adaptive.”
|
||
Response to Comments on the Quality of English Language |
||
Point 1: Very minor grammar finds: lines 389 and 390 switch into a different verb tense rather than the past tense. “extended”, “found” “were”.
Great, thanks. We have now fixed those grammar errors. The fixed text appears on lines 434-439 and reads: Following calls from recent works, we explored themes in narratives told with a redemptive structure (Perlin & Fivush, 2021). We extended those calls by analyzing the themes by nation. We found that US participants were most likely to tell redemptive stories that focus on personal growth of the self, whereas UK participants did not elaborate deeply on how they grew or they told stories of mastery that expressed their sense of competence and autonomy (see Figure 2).
|
Reviewer 3 Report
Comments and Suggestions for Authors
Review of ‘A Comparison of Turning Point Memories among US and UK Emerging Adults: Adversity, Redemption, and Unresolved Trauma.’
This was a mixed methods study that assessed ‘turning point’ autobiographical memory narratives from college-aged students in the United States and the United Kingdom and whether those narratives predicted changes in well-being and emotions.
The authors describe their samples as emerging adults, although the mean age of both samples are ~20 years, which may be more specifically defined as adolescent samples. Further, when doing ‘cross-cultural’ work, the variability within cultures is probably more pronounced than the variability between cultures. For example, a sample of ‘Mountain West’ adolescents may show less variability than a smaller sample of regional university adolescents who identified as citizens of ‘England…Bulgaria/Greece, etc.’.
Pre- post measures of Psychological Well-Being Scale, the PANAS positive and negative affect/mood scale. Coders coded redemptive narration.
Under 3.2c section there were no differences between cultures on redemptive narration, so there is no need to report a mean difference. The lack of differences in the authors’ correlations with redemption and well-being and emotions between culture samples is explained away by lack of UK sample size. Would it be difficult to generate an equal sized sample in the UK compared to the US sample to test these again?
The discussion mentions resolution (open/closed) memories but needs to reference the studies that have coded psychological resolution, and why that might be important in redemption understanding.
The authors allude to not completely capturing narratives of emerging adults in their findings, and it would be beneficial to explain to readers that in future research, including non-college aged persons in their 20s through 30s would be more enlightening in this area of research.
This reviewer would like to see a rewritten version incorporating some of the areas of comment above before accepting the manuscript.
Author Response
A Comparison of Turning Point Memories among US and UK Emerging Adults: Adversity, Redemption, and Unresolved Trauma
Response to Reviewer 3’s Comments
|
||
Summary |
|
|
Thank you very much for your helpful comment and for taking the time to review our manuscript. We feel like your comments were particularly helpful in improving our work. Below, we provide your original comments (italicized in black) starting with the summary statement and any corresponding changes based on those comments (regular font in red). We include the page number, and the lines on which the changes appear in the revised manuscript.
Reviewer 3 Summary Statement This was a mixed methods study that assessed ‘turning point’ autobiographical memory narratives from college-aged students in the United States and the United Kingdom and whether those narratives predicted changes in well-being and emotions. The authors describe their samples as emerging adults, although the mean age of both samples are ~20 years, which may be more specifically defined as adolescent samples. Further, when doing ‘cross-cultural’ work, the variability within cultures is probably more pronounced than the variability between cultures. For example, a sample of ‘Mountain West’ adolescents may show less variability than a smaller sample of regional university adolescents who identified as citizens of ‘England…Bulgaria/Greece, etc.’. Pre- post measures of Psychological Well-Being Scale, the PANAS positive and negative affect/mood scale. Coders coded redemptive narration.
|
||
Point-by-point response to Comments and Suggestions for Authors |
||
Comment 1: Under 3.2c section there were no differences between cultures on redemptive narration, so there is no need to report a mean difference. |
||
Response 1: Thank you for this comment. We have revised that section. It now reads (lines 442-447, page 10): “Do people in the two samples tell redemptive narratives with similar frequency? Our next question focused on redemptive narration. Are people from the two samples similarly likely to move from negative to a positive ending in their turning point memory narratives? Independent samples t-test with 1,000 bootstrapped re-estimates showed that US and UK participants did not significantly differ on redemptive narration (see Table 1).”
|
||
Comment 2: The lack of differences in the authors’ correlations with redemption and well-being and emotions between culture samples is explained away by lack of UK sample size. Would it be difficult to generate an equal sized sample in the UK compared to the US sample to test these again?
|
||
Response 2: Agree. Based on your comment and another reviewer’s comment we now explain why our UK sample was so small in relation to the US sample. We are likely to try to resample in the UK in future work with a modified protocol, study design, and research questions. Thanks for your understanding. At any rate, we now state the following On Line 244 – 247, page 6: “UK data collection was part of a student research project that was part of a study abroad experience. We made numerous efforts to recruit our sample and re-recruit but we had low participation rates overall. When the study abroad experience ended, we ceased data collection. This is discussed more in the study limitations.” To help readers have a better sense of the analyses given the unbalanced sample sizes we also added the following in a note below Table 1 (where the comparisons occur). That note reads: Required sample size for a difference that reaches a medium effect size (d = .05) with alpha =.05 and power at .8 G*power 3.1 indicates we would need a sample size of 208 with as few as 39 UK participants and 169 US participants. As an example of statistical power we actually achieved, post-hoc achieved power for effect size .69 with alpha = .05, and n = 25 (UK) and n = 107 (US) power = .87 according to G*power 3.1.
Comment 3: The discussion mentions resolution (open/closed) memories but needs to reference the studies that have coded psychological resolution, and why that might be important in redemption understanding.
Response 3: Thank you for pointing this out. We agree with this comment and it aligns with a comment from another reviewer about improving the discussion by returning to the Perlin and Fivush (2021) paper. Because of these comments we have greatly expanded the discussion and we include a new reference to a publication in which we coded for narrative resolution. Lines 639-693 on pages 15 through 16 now reads as: Whether or not one can live with the trauma and narrate it in the life story in a psychologically adaptive way without positively storying that memory is an important question. Perhaps there is a way that people can accept and live with trauma without being defined by it that is psychologically beneficial that we have not measured in our studies. This idea is perhaps at the core of the potential benefits of recuperative narration (Blackie et al., 2023), return identity themes in life stories (Perlin & Fivush, 2021) and “post” post-traumatic growth (Adler & Schwaba, 2024). These questions are especially relevant now as narrative studies are beginning to take account of cultural variability in meaning-making. Perlin and Fivush’s model may be especially useful in future work on cultural-situated meaning-making because its central focus is variability in how people express resolution. Indeed, they argue that “by attending to the situation and identity themes present across the entire narrative, we move beyond the specific forms of redemption that are culturally, religiously, and developmentally constrained (Perlin & Fivush, 2021, p. 28).” Perlin & Fivush (2021) propose a contextualized, lifespan model of redemptive narration of adversity. They argue that there may be age-graded differences in how redemptive themes are manifest in adversity narratives and in the function redemptive themes serve. Their model begins with the type of adversity. They argue that we can have situation-specific adversity, such as failing a class in college, and identity-relevant adversity, such as struggling to choose a career path that makes sense for the self broadly. They go on to specify two forms of redemptive resolution (for other published works on resolution of negative experiences see Mansfield et al., 2010). One, return resolution, creates stability for the self by expressing a return of functioning, or capability, or capacity that was challenged by the adversity. The second, emergent resolution, signifies the development of a new strength, the identification of a novel skill, or a new insight about the self as a result of experiencing the adversity (see also Pasupathi, et al., 2007 for seminal theory on self-event connections in personal narratives). Hence in this model the way that resolution is expressed is critical for a nuanced understanding of redemptive narratives because it goes beyond quantitative approaches that show whether or not and the degree to which the individual redeemed to provide the quality of redemptive resolutions. Toward an even stronger refinement of redemptive narration theory, they suggest that return and emergent forms can express situation or identity themes. Taken as a whole this is an elegant and flexible model that we feel explains our turning point data well. It suggests, for example, that a relatively minor situational adversity (e.g., failing a class) can be resolved with a return and identity relevant theme such as rediscovering a love of learning that the narrator expresses as a core component of identity. Although we did not code our narratives in the same terms that Perlin and Fivush used, we recognize many of these types of stories in our turning point dataset. In both the US and UK, broad themes of mastery, personal growth, and positive relatedness to others started out as situational adversity (for example, see the narrative about traveling in a new country with friends) but bloomed into stories of deeper identity concerns. That makes sense given that we prompted for turning points, which are identity relevant and our participants are emerging adults for whom identity development is important. Others began with an identity-relevant situation and resolved it with identity relevant themes, such as the example above about being diagnosed with Type 1 Diabetes. Yet overall, our work points to a tension in the progression of redemptive narrative research (see also Mansfield et al., 2025). For example, McLean and colleagues (2020) coded redemption as a structural element and argued that conceptually redemption has elements of both autobiographical reasoning (thinking about and making connections between self and experiences in narratives) and motivational and affective themes. Likewise, we coded redemption as a structural feature in our study and examined the themes in our narratives. We believe that this is valuable because it allows both quantitative and qualitative understanding of the data fitting the coding to the conceptualization. Further, it may be that the eudaemonic and hedonic benefits of redemptive telling arises out of a combination of structure (ending on a positive note), theme, and reasoning. As of now this is an area for ongoing research (Mansfield et al., 2023).
Comment 4: The authors allude to not completely capturing narratives of emerging adults in their findings, and it would be beneficial to explain to readers that in future research, including non-college aged persons in their 20s through 30s would be more enlightening in this area of research.
Response 4: Thank you for pointing this out. We agree with this comment. Therefore, we have extended the information in the limitations section (lines 703-707, page 16) to say: In both samples, our data came from emerging adults attending college, who may not be representative of emerging adults more generally. Indeed, studying narrative identity development through the 30’s in people who have neither attended nor completed 2- or 4-year degrees would be an illuminating area of future work.
|